

# Autonomous wave gliders as a tool to characterize delphinid habitats along the Florida Atlantic coast

Jessica Carvalho, Laurent M. Chérubin and Greg O'Corry-Crowe

Florida Atlantic University (Harbor Branch Campus), Fort Pierce, FL, United States of America

## ABSTRACT

As climate change and anthropogenic activities continue to impact cetacean species, it becomes increasingly urgent to efficiently monitor cetacean populations. Continuing technological advances enable innovative research methodologies which broaden monitoring approaches. In our study, we utilized an autonomous wave glider equipped with acoustic and environmental sensors to assess delphinid species presence on the east Florida shelf and compared this approach with traditional marine mammal monitoring methods. Acoustic recordings were analyzed to detect delphinid presence along the glider track in conjunction with subsurface environmental variables such as temperature, salinity, current velocity, and chlorophyll-a concentration. Additionally, occurrences of soniferous fish and anthropogenic noise were also documented. These *in-situ* variables were incorporated into generalized additive models (GAMs) to identify predictors of delphinid presence. The top-performing GAM found that location, sound pressure level (SPL), temperature, and chlorophyll-a concentration explained 50.8% of the deviance in the dataset. The use of satellite environmental variables with the absence of acoustic variables found that location, derived current speed and heading, and chlorophyll-a explained 44.8% of deviance in the dataset. Our research reveals the explanatory power of acoustic variables, measurable with autonomous platforms such as wave gliders, in delphinid presence drivers and habitat characterization.

## INTRODUCTION

Cetaceans are widely distributed in Florida waters, from shallow coastal areas to the deep offshore environment, playing a critical role in ecosystem function as upper trophic level species (*Bowen, 1997*; *Kiszka, Woodstock & Heithaus, 2022*). Bottlenose dolphins (*Tursiops truncatus*) inhabit nearshore waters, including bays, estuaries, and the continental shelf year-round (*Mazzoil et al., 2011*; *Mazzoil et al., 2020*), while species such as Atlantic spotted dolphins (*Stenella frontalis*), short-finned pilot whales (*Globicephala macrorhynchus*), and Risso's dolphins (*Grampus griseus*) are more commonly found offshore (*Griffin & Griffin, 2004*; *Herzing & Elliser, 2016*). Seasonal visitors like North Atlantic right whales (*Eubalaena glacialis*) use Florida's Atlantic waters as critical calving grounds during winter months (*Keller et al., 2006*; *Garrison et al., 2012*). As upper trophic level species, including some

Corresponding author
Jessica Carvalho, jcarvalho@fau.edu

apex predators, these marine mammals play a critical role in ecosystem function (*Bowen, 1997*; *Kiszka, Woodstock & Heithaus, 2022*), emphasizing the importance of understanding their distribution and habitat utilization for effective management and conservation.

Historically, efforts to monitor cetacean populations have relied heavily on visual shipboard and aerial surveys or opportunistic sightings (*Di Tullio et al., 2016*; *Herzing & Elliser, 2016*; *Roberts et al., 2016*; *Garrison & Dias, 2023*). While these methods can facilitate species identification, behavioral assessment, and in some cases, population abundance estimates (*Griffin & Griffin, 2004*; *Elliser & Herzing, 2012*; *Herzing & Elliser, 2016*; *Haria et al., 2023*), they also have considerable drawbacks, such as high costs and susceptibility to inclement weather. Research in the coastal and continental shelf waters off Florida's east coast has provided insights into habitat suitability and population density for several delphinid species and right whales (*Herzing & Elliser, 2016*; *Roberts et al., 2016*; *Chavez-Rosales et al., 2019*; *Chavez-Rosales, Josephson & Garrison, 2022*). However, these studies have primarily relied on visual observations and contain limited contemporary data, with a notable scarcity of observations in our specific study area (*Rickard, 2015*). Given the high costs of visual surveys, their susceptibility to weather conditions, and challenges in detecting species with prolonged dive durations, alternative approaches are necessary to improve data collection and coverage.

The incorporation of environmental predictors such as sea surface temperature (SST), salinity, bottom temperature among others into models of cetacean presence and density has been shown to enhance our understanding of cetacean habitat selection (*Forney, 2000*; *Chavez-Rosales et al., 2019*). For example, SST can distinguish thermal boundaries, and depth can establish habitat partitioning (*Forney, 2000*; *Chavez-Rosales et al., 2019*). Identifying and analyzing these environmental variables can help researchers better delineate important habitats, predict cetacean presence, and inform the design of future research and conservation strategies. Yet, traditional approaches often overlook these factors, and when included, rely on satellite data for environmental variables which can introduce constraints due to the lack of fine-scale resolution and sub-surface information compared to *in situ* data sources, susceptibility to cloud coverage, and synopticity. As such, there is a pressing need to explore innovative and cost-effective monitoring strategies that also continuously measure environmental variables while tracking the presence of the animals.

Recognizing the inherent limitations in traditional survey methods, there has been an increasing shift toward acoustic monitoring methods, such as stationary passive acoustic monitoring (PAM) (*Wiggins & Hildebrand, 2007*; *Sousa-Lima et al., 2013*; *Malinka et al., 2018*; *Rafter et al., 2021*) and autonomous platforms equipped with acoustic recorders for monitoring of cetacean species (*Klinck et al., 2012*; *Klinck et al., 2016*; *Bittencourt et al., 2017*; *Bittencourt et al., 2018*). Acoustic monitoring offers a less invasive and more cost-effective alternative to traditional survey methods, allowing for extended deployments and improving detectability of species with long dive times. Unlike vessels, which may alter cetacean behaviors (*Richardson & Würsig, 1997*; *Hawkins & Gartside, 2009*; *Bas et al., 2017*), acoustic monitoring systems, particularly stationary PAM, are quieter and less likely to interfere with natural behaviors. These systems can operate continuously 24 h a day, 7

days a week, even in inclement weather conditions, without the need for human presence in the field. Stationary hydrophones provide valuable data for long-term monitoring, capturing vocalizations over extended periods, which is crucial for understanding diel patterns and seasonal migrations. PAM also enables the assessment of changes in the soundscape associated with the habitat spatial and temporal variability. Changes to the ocean soundscapes caused by anthropogenic activities (*e.g.*, shipping, oil, and gas exploration) may not only impact cetaceans (*Weilgart, 2007*; *Martin et al., 2023*), but also affect the presence of soniferous fish species (*Simpson et al., 2016*; *De Jong et al., 2020*), some of which may be prey species (*Barros & Wells, 1998*; *Gannon & Waples, 2004*). The life history of marine animals, including mammals, fish, and invertebrates can be impacted by each species' perception of various sound sources and how an individual animal may rely on sound to forage, communicate, and/or avoid predators (*Erbe et al., 2019*; *Popper & Hawkins, 2019*; *Haver et al., 2023*). *Jensen et al. (2009)* showed that vessel noise can impact delphinids communication in shallow water. They found that bottlenose dolphins (*Tursiops* sp.) could suffer a reduction in their communication range of 26% within 50 m. Effects on the movement, behavior and vocalizations of bottlenose dolphins were also identified in *Marley et al (2017)*. PAM can enhance detection capabilities for cetacean species with extended dive durations (*i.e.*, beaked whales, *Hildebrand et al. (2015)*), offering the potential for new ecological insights such as preferred habitat and related activity.

Stationary PAM has been used to monitor species like the Risso's dolphin along the California coast and Antarctic blue whales (*Balaenoptera musculus*) (*Soldevilla, Wiggins & Hildebrand, 2010*; *Thomisch et al., 2016*). These systems typically do not measure environmental variables, although sea surface conditions are accessible through satellite measurements. The limiting factor with stationary systems is the limited acoustic detection range which necessitates multiple spatially distributed units for broader coverage. In contrast, autonomous vehicles equipped with acoustic recorders offer a more dynamic approach, enabling cetacean monitoring over larger spatial scales. These tools have been used to monitor beaked whales off the coast of Hawai'i, delphinid species off the coast of Brazil and several species of baleen whales in the Gulf of Maine (*Klinck et al., 2012*; *Baumgartner et al., 2013*; *Bittencourt et al., 2017*). These platforms can traverse vast ocean areas, enabling the monitoring of a broader range of cetacean species, from coastal to deep-diving species, such as beaked whales and sperm whales, across continental shelf and slope waters (*Klinck et al., 2012*; *Bittencourt et al., 2017*; *Bittencourt et al., 2018*). Autonomous platforms not only capture cetacean vocalizations but are also often equipped with additional sensors to measure environmental variables, such as temperature, salinity, and chlorophyll-a concentration, which can help researchers understand habitat preferences and distribution patterns of the monitored species. This combination of sensors may prove particularly informative in regions where fixed hydrophones might be impractical. The ability to monitor cetaceans over broad spatial scales with mobile platforms complements the long-term data gathered by stationary systems, providing a comprehensive picture of cetacean presence and behavior in various ocean environments.

In this study, we describe a wave glider survey that took place on the east Florida shelf. The wave glider carried a PAM system, an environmental sensors payload, a CTD, and a

current velocity profiler that enabled simultaneous habitat and soundscape monitoring associated with the habitat spatial and temporal variability. The collected data reveals the association of environmental variables, biological, and anthropogenic noise in explaining delphinid presence using generalized additive models (GAMs). We ultimately demonstrate that the addition of the acoustic variables improves the explanatory performance of the model, more than the use of *in-situ versus* satellite measured environmental variables.

## METHODS

### Data processing

In March 2019, a Liquid Robotics autonomous wave glider (WG) was deployed for a two-month long survey on the east Florida shelf between Fort Pierce and Jacksonville, Florida. The WG survey consisted of successive across-shelf transects from the 10 m isobath to the edge of the Gulf Stream (Fig. 1). The WG which belongs to the autonomous surface vehicle category (see *Chérubin et al. (2020)* for further details), is wave propelled and consists of a surface float with solar panels tethered to a submersible glider by an umbilical cable. The surface float houses the electronics, power, lights, surface current sensor and communication getaways, including user-specified payload. The submersible glider acts as the propulsion mechanism and is connected to a custom-built tow-body equipped with acoustic and environmental sensors (Fig. 2). The tow-body is ballasted to be neutrally buoyant between four and ten m below the surface float and trails ten m behind the submersible glider. A web-based interface, called WGMS is used to navigate, communicate and access in real-time glider and sensor data during missions. Communication with the WG is done through cellular network or Iridium satellite.

During the 2019 survey, the glider measured environmental variables such as surface current speed and direction, and velocity profiles down to 50 m using a 600 kHz Workhorse ADCP mounted on the surface float. The tow-body was equipped with a self-powered, self-logging EXO1 YSI multiparameter sonde which collected pressure, pH, temperature, salinity, and dissolved oxygen data. Additionally, a Turner C3 Fluorometer measured CDOM, chlorophyll-a, and backscattering fluorescence. Acoustic recordings were obtained with a standalone Remora-ST from Ocean Instruments mounted on the tow-body (Fig. 2), following a duty cycle of 15 s every five minutes. This duty cycle enables a two-month long battery life, the Remora-ST recorded continuously over a 54-day period from 8 March 2019 until 30 April 2019. Recordings were made at a sampling rate of 48 kHz with pre-amp gain that results in a hydrophone sensitivity of −176 dB re: 1V/μPa. Additionally, the glider operated a real-time acoustic monitoring system designed for low frequency sound recording with a sampling rate of 10 kHz, a hydrophone sensitivity of −201 dB re: 1 V/μPa without pre-amp. Recordings were made for 30 s every minute. This passive acoustic monitoring system was mostly used for the assessment of fish presence and provides to the glider user real-time access to the audio recordings. The low sampling frequency enables the storage of a larger amount of acoustic data files on the glider and the possibility of transferring acoustic files through cellular or over Iridium at reasonable costs.

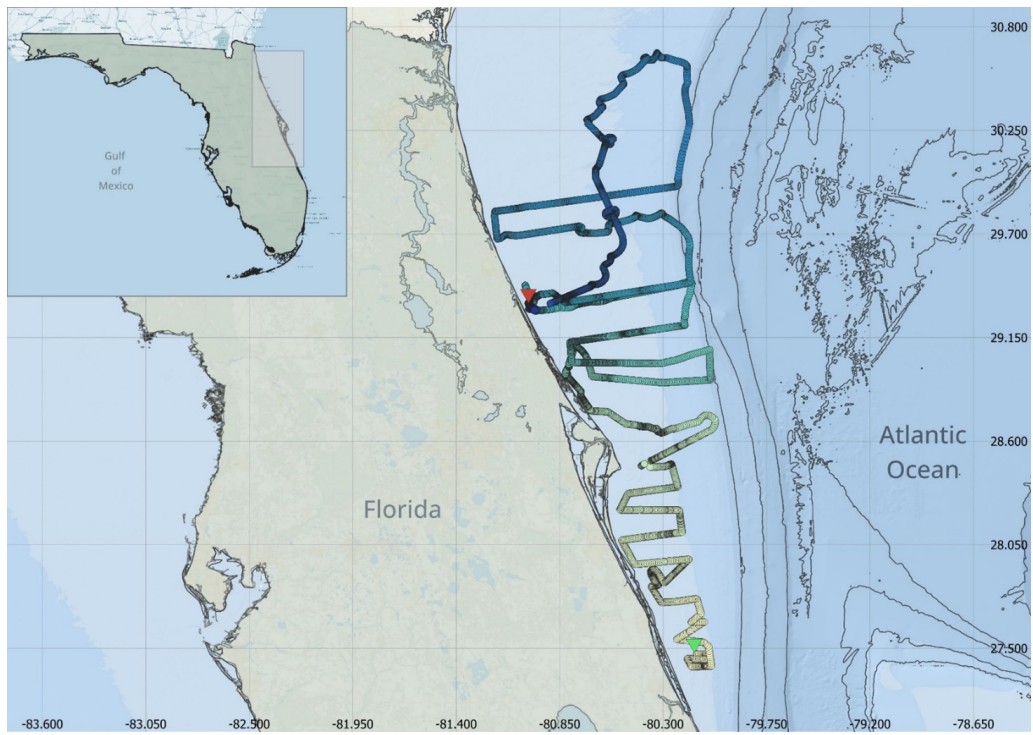

**Figure 1 Glider track along the Florida Atlantic coast.** The glider track is colored by date with the start date (indicated by the green arrow) represented with the lightest color and the end date (indicated by the red arrow) represented with the darkest color. Bathymetry is represented by grey solid contour lines. The transition from the continental shelf to the slope is demarcated by the first 200 m isobath contour. The contour interval is 200 m.

## Acoustic data processing

Acoustic files from the Remora-ST were reviewed to assess the presence of delphinids in the mid/high frequency bands using spectrograms generated in Raven Pro 1.6 with a 1024-point DFT and 60% overlap. A single observer (JC) visually reviewed the recordings in their entirety. Points of interest identified visually, were further subjected to auditory review (JC) to ensure annotation accuracy. Due to frequent vocalization overlaps, signals were not individually selected. Instead, each 15-second file was categorized as either containing detected signals (presence = 1) or not (absence = 0) based on the detection of delphinid signals. During the glider deployment, the Remora recorded a total of 14,974 acoustic files, amounting to approximately 62 h of recording. Delphinids were detected in 11% ($n = 1,694$) of these recordings, with whistles accounting for 55.3% of detections, echolocation clicks for 29.4%, and 15.3% contained both whistles and echolocation clicks. Other sounds such as buzzes and burst pulses were also accounted for if present, alongside whistles and echolocation clicks within each 15-second file.

## Habitat variables and spatio-temporal mapping

Environmental variables measured by the WG and used in this study included subsurface salinity, temperature, surface current speed and heading from the glider navigation

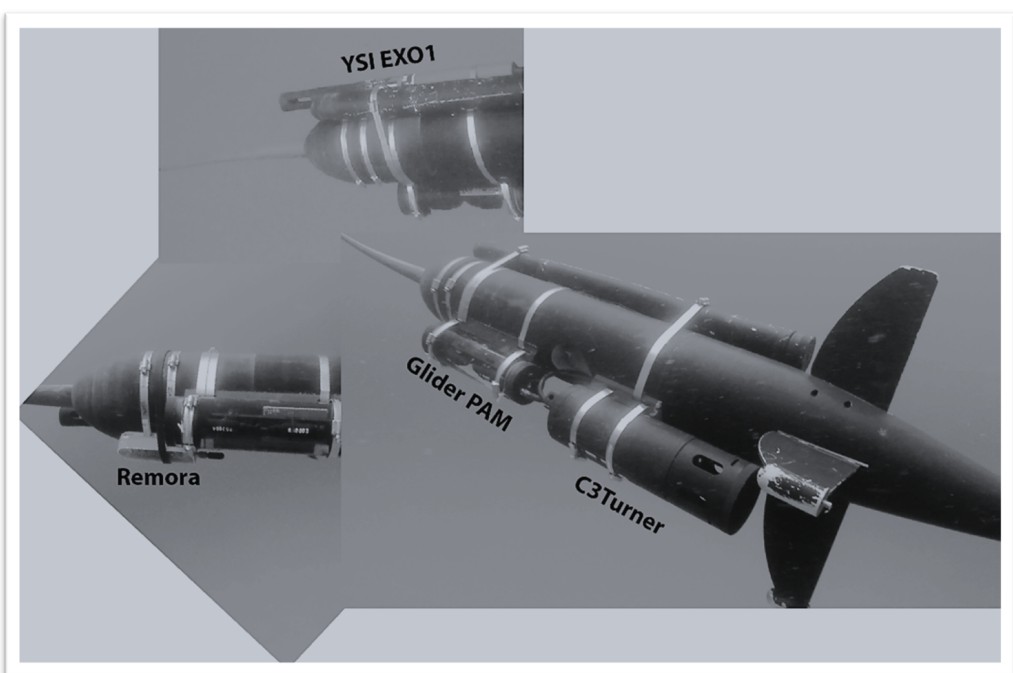

**Figure 2   Glider sensor payload on the tow-body.** Images of the glider sensor payload on the tow-body during the 2019 east Florida shelf survey showing the Remora acoustic recorder, the YSI EXO1 CTD, the glider passive acoustic monitoring system (PAM) and the C3 Turner fluorometer.

system, and chlorophyll-a concentration from the Turner C3 (Table 1). The chlorophyll-a measurements contained several large data gaps, when measurements ceased to be recorded. To account for these gaps which would have made the data inadequate for this study, a generalized additive model was used to assess the relationship between the *in-situ* and satellite measurements. The predictive model was then used to fill in the gaps in the glider chlorophyll data.

Satellite derived estimates of current speed and direction, and chlorophyll-a, were obtained from Copernicus Marine Service (*Copernicus Marine Data Store, 2020a*; *Copernicus Marine Data Store, 2020b*). SST was retrieved from NOAA Coast Watch (*NOAA CoastWatch, 1997*). Satellite datasets were obtained with a start and end time matching the period of the glider deployment and the measurement value of the pixel within which the WG was located was assigned to the glider location. Further information on the satellite datasets including data sources, web access and spatial resolution can be found in Table 1.

The spatio-temporal representation of the data included location parameters (date and location ID) and the distance to the coast, calculated using the glider track. Depth data along the glider track was sourced exclusively from the online database Gebco (*GEBCO, 2023*). Soundscape metrics included sound pressure level (SPL), the acoustic presence of fish, and anthropogenic noise presence. Sound pressure levels (SPL) were averaged over the 300–5,000 Hz frequency range for each file using PAMGuide (*Merchant et al., 2015*), an acoustic analysis software in MATLAB (*Mathworks, 2022*). This frequency range was

**Table 1  All predictor variables selected for GAM analysis of cetacean presence.** Abbreviations are included for every variable, along with a description of the variable measured, the resolution at which the data was recorded, and the origination source. Variables with multiple descriptions were acquired from multiple sources and are representative of either glider or satellite data. This distinction was made in order to capture the differences in resolution between these two sources.

| Abbreviation | Description | Resolution | Source |
|---|---|---|---|
| Ant | Anthropogenic noise (presence/absence) | *In Situ* | Glider remora |
| SPL | Sound pressure level 300 Hz–5,000 Hz | *In Situ* | Glider remora |
| Fish | Fish (presence/absence) | *In Situ* | Glider remora |
| Dist | Distance to the coast (m) | | QGIS |
| Sal | Salinity (ppt) | *In Situ* | Glider YSI |
| | Salinity (ppt) | 0.083° × 0.083° | https://doi.org/10.48670/moi-00021 |
| Temp | Sub surface temperature (°C) | *In Situ* | Glider YSI |
| | Sea Surface Temperature (°C) | 1 × 1 km | https://coastwatch.noaa.gov/erddap/griddap/noaacwecnMURdaily.html |
| Dep | Bathymetry (m) | | https://download.gebco.net/ |
| Dist | Distance to the coast (m) | | QGIS |
| CurS | Current speed (m/s) | *In Situ* | Glider |
| | Current speed (m/s) calculated from eastward and northward water velocities | 0.083° × 0.083° | https://doi.org/10.48670/moi-00021 |
| CurH | Current heading (Degrees) | *In Situ* | Glider |
| | Current heading (Degrees) | 0.083° × 0.083° | https://doi.org/10.48670/moi-00021 |
| Chla | Concentration of chlorophyll-a (RFU (raw fluorescence units)) | *In Situ* | Glider C3 Turner |
| | Mass concentration of chlorophyll a in sea water (mg/m³) | 4 × 4 km | https://doi.org/10.48670/moi-00281 |

selected to capture the lower range of SPL in the natural soundscape (*e.g.*, wind and wave action) and the general range of anthropogenic noise while minimizing the influence of self-noise from the glider, which was occasionally produced below 300 Hz. In addition, the presence of anthropogenic noise sources (particularly vessel noise and occasional sonar) and soniferous fish (including toadfish *(Opsanus sp.)*, jacks *(Caranx sp.)*, and drums *(Pogonias sp.)*) known as potential preys (*Barros & Wells, 1998*; *Gannon & Waples, 2004*) were identified both visually and aurally, by one observer (JC), and categorized in the same manner as delphinid presence, as either present or absent. The same spectrogram settings were used for all detections; however, fish sounds were identified in the 0–1,000 Hz band and vessel and sonar noise in the 0–6 kHz. Those sound sources were counted only when present in both the spectrograms and aurally. Noise from the glider is clearly identifiable and was discarded and at times overlapped with the background noise. In that case, this part of the recording was not used in the analysis. All variables were daily averaged for consistency with the temporal resolution of the satellite data.

Delphinid signals, particularly whistles, can be detected over large distances, sometimes exceeding 20 km from the signal-producing individual (*Quintana-Rizzo, Mann & Wells, 2006*). However, the effectiveness of sound propagation of whistle-like sounds by dolphins and their detectability distances have been found to vary substantially with habitat characteristics (*Quintana-Rizzo, Mann & Wells, 2006*). To account for the possibility of delphinid presence within a circle of 10 km radius around the WG, given the uncertainty

on proximity to the WG, the dataset was averaged over a 10 km hexagonal grid. All spatial analyses were conducted using QGIS (version 3.36.3) and RStudio (version 4.3.2).

## Statistical analysis

Generalized additive models (GAMs) with a Tweedie distribution were used to assess the influence of various factors on delphinid presence. The Tweedie distribution was selected for the GAM models based on its demonstrated effectiveness in handling sparsely distributed species (*Miller et al., 2013*). Separate models were developed with either exclusively glider data or exclusively satellite data, to compare the efficacy of gliders against traditional methods. The glider-only model (GOM) included the variables: location (date, ID), anthropogenic noise, SPL, fish presence, distance to the coast, salinity, temperature, water depth, surface current speed and heading, and chlorophyll-a concentration. The satellite-only model (SOM) included location, distance to the coast, salinity, sea surface temperature, water depth, derived current speed and heading, and chlorophyll-a concentration. Soundscape variables of anthropogenic noise, fish presence, and SPL were not included in the SOM as these variables cannot be derived from satellite sources and are not typically collected in traditional marine mammal surveys. All GAMs were run in RStudio (version 4.3.2).

After running the initial GAM models with all variables, the dredge function from the MuMIn package in RStudio (version 4.3.2) was employed to identify the most effective models based on Akaike Information Criterion (AIC) scores. This process produced a comprehensive list of models along with their respective AIC scores, facilitating the selection of the best-fitting models as indicated by the lowest AIC values.

## RESULTS

The results of GAMs, as shown in Table 2, highlight the top three performing models for both the GOM and the SOM. The best-fitting GOM, explaining delphinid presence, included four variables: SPL, location (date, location ID), temperature, and chlorophyll-a concentration. This model explained 50.8% of the deviance in the dataset with an AIC score of -61.05 and an R-squared value of 0.501. All variables in this model were statistically significant. The complexity of each smooth term, represented by the estimated degrees of freedom (edf), along with reference degrees of freedom (Ref.df; available degrees of freedom), F-statistic (F; contribution to the model), and *P*-value (statistical significance), are detailed in Table S1. Smooth plots showing the relationships between predictor variables and delphinid presence can be seen in Fig. 3. The analysis revealed a non-linear relationship between odontocete presence and SPL, indicating a general negative association at higher decibels. There is, however, an increase in predicted delphinid presence at mid and again at high SPL levels. Delphinid presence was positively correlated with chlorophyll-a, although the relationship is also non-linear. Delphinid presence seemed to increase with chlorophyll-a although some avoidance to mid/high chlorophyll-a concentration areas is suggested. The variable with the narrowest confidence interval was temperature, and it suggests that delphinid presence was negatively correlated with increasing sub-surface temperature. The relationship between delphinid presence and location (Fig. 4) indicated

**Table 2  Comparison of the top three best-fit GAM models from the glider-only data model (GOM) and satellite-only data model (SOM).** Variables included in the model are represented with (+). Akaike information criterion (AIC) scores are included for model comparison, as well as adjusted R-squared values (R-sq) and percentage of deviance explained (Dev). Also included are the top three models of the GOM without sound variables.

| | Ant | SPL | Fish | Location | Dist | Sal | Temp | Dep | CurS | CurH | Chla | AIC | R-sq | Dev |
|---|---|---|---|---|---|---|---|---|---|---|---|---|---|---|
| Glider only model | | + | | + | | | + | | | | + | −61.05 | 0.501 | 50.8% |
| | | + | | + | | | + | | + | | + | −60.90 | 0.500 | 51.7% |
| | | + | | + | | | + | + | | | + | −60.53 | 0.506 | 51.5% |

| | | | | Location | Dist | Sal | SST | Dep | CurS | CurH | Chla | AIC | R-sq | Dev |
|---|---|---|---|---|---|---|---|---|---|---|---|---|---|---|
| Satellite only model | | | | + | | | | | + | + | + | −34.19 | 0.461 | 44.8% |
| | | | | | | + | + | + | + | + | + | −31.73 | 0.477 | 44.9% |
| | | | | + | | | + | | + | + | + | −31.90 | 0.458 | 44.8% |

| | | | | Location | Dist | Sal | Temp | Dep | CurS | CurH | Chla | AIC | R-sq | Dev |
|---|---|---|---|---|---|---|---|---|---|---|---|---|---|---|
| Glider only model no sound | | | | + | | | + | | | | + | −47.62 | 0.468 | 47.0% |
| | | | | + | | | + | + | + | + | + | −47.47 | 0.461 | 49.2% |
| | | | | + | | | + | | + | | + | −47.30 | 0.464 | 47.9% |

an increase in presence closer to shore, with a few concentrated areas of increased presence spanning south of Cape Canaveral to the north rather than the southern portion of the survey. The highest presence was observed near St. Augustine Inlet and the second highest southwest of Cape Canaveral. It is also essential to note the 95% confidence intervals for these variables, which indicate a level of uncertainty within certain ranges of the variables.

The top-performing SOM contained four variables: location, current speed, current heading, and chlorophyll-a concentration (Table 2). This model explained 44.8% of the deviance in the dataset with an AIC score -34.19 and an R-squared value of 0.461. All variables in this model were statistically significant except current heading. *P*-values, edf, Ref.df, and the F-statistic for each variable are provided in Table S1. Smooth plots of the resulting top model are depicted in Fig. 5. Results show that delphinid presence is positively correlated with chlorophyll-a. Delphinid presence showed a negative correlation with south eastward currents, where the uncertainty level was the lowest. Furthermore, delphinid presence was negatively correlated with current speed. The relationship between delphinid presence and location, showed an increase in presence closer to shore, identical to the GOM (Fig. 6).

For a true comparison between glider and satellite data, an additional dredge function was run on *in-situ* data without the acoustic variables. The variables included were: location, distance to the coast, salinity, sub-surface temperature, depth, current speed and heading, and chlorophyll-a concentration. The resulting top performing model included location, temperature, and chl-a, which explained 47.0% of the deviance in the dataset, with an AIC of -47.62 and an R-squared value of 0.468. This model has a lower explained deviance than the GOM, but similar deviance to the SOM model.

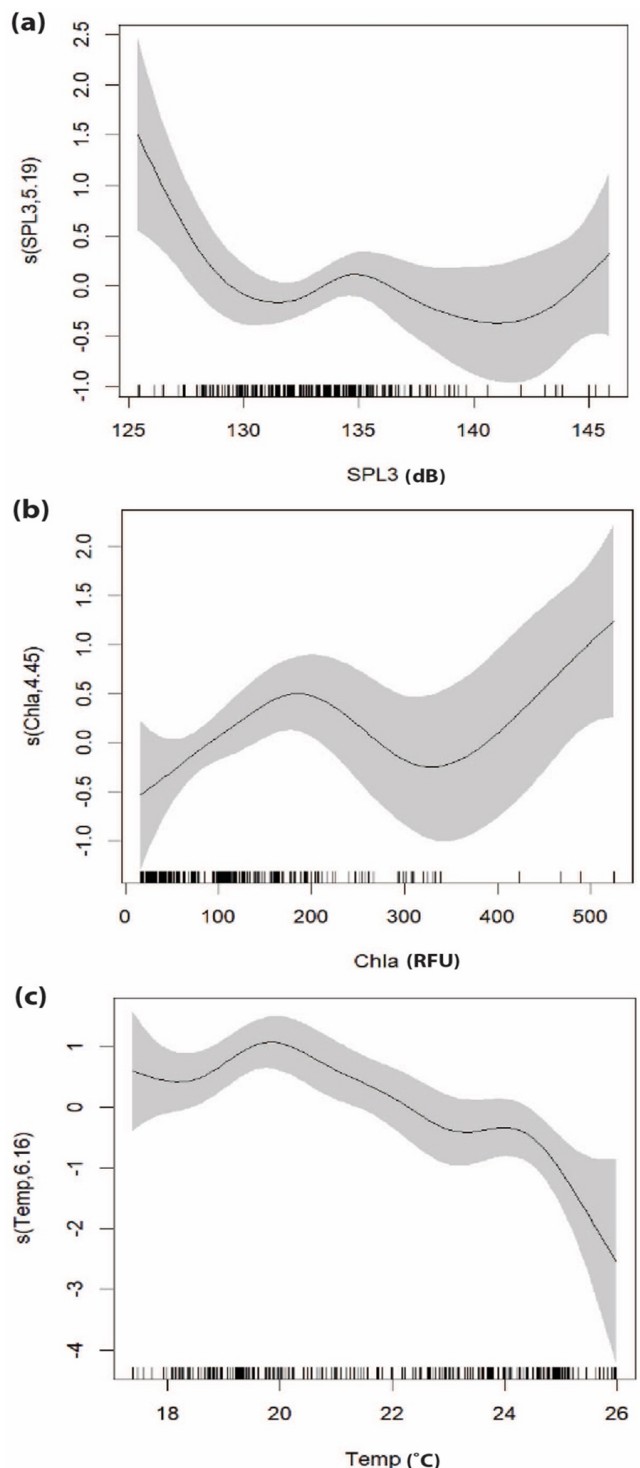

**Figure 3  Smooth plots displaying the results of the top-performing GOM assessing the relationship between delphinid presence and environmental variables.** (A) SPL (dB), (B) chlorophyll-a (RFU), and (C) temperature (°C). Tick marks on the $x$-axis represent data observations and gray shading represents the 95% confidence interval. The $y$-axis represents the relative (continued on next page…)

**Figure 3 (…continued)**
effect of each variable on the presence of delphinids, with values above zero indicating increased presence and numbers below zero representing decreased presence. The number associated with each variable on the *y*-axis represents the effective degrees of freedom (EDF) for each smooth term, indicating the complexity of each variable's relationship with delphinid presence, 1 being the least complex.

## DISCUSSION

As anthropogenic impacts on the ocean continue to increase (*Halpern et al., 2007*; *Halpern et al., 2008*; *Häder et al., 2020*), understanding the distribution of cetacean species, their habitat preferences and the human activities that may be displacing them becomes increasingly important for informed management. Our study contributes to the existing knowledge of cetacean habitat use (*Garrison, Martinnez & Maze-Foley, 2010*; *Best et al., 2012*; *Roberts et al., 2016*) by utilizing an autonomous WG for odontocete monitoring. Mostly delphinid species were identified in our dataset but species-specific identification was limited. Many Atlantic delphinid sounds are not yet distinguishable to species based on the character of their clicks, buzz or burst pulses, or whistles (*Roch et al., 2011*; *Gillespie et al., 2013*). For instance, common dolphin species (short-beaked and long-beaked) and bottlenose dolphins make clicks that are thus far indistinguishable from each other (*Soldevilla et al., 2008*). Risso's dolphins were occasionally detected based on known characteristics (*Soldevilla et al., 2008*), and blackfish sounds were suspected. However, the Atlantic shelf of Florida is home to a large number of dolphin species as suggested by *Rafter et al. (2021)*.

Despite the lack of identification at the species level, our autonomous vehicle approach overcomes several limitations associated with traditional survey methods such as high costs, limited survey durations and track lengths, susceptibility to weather conditions, and challenges in detecting species with prolonged dive durations. Additionally, we evaluated the effectiveness of different variables and their sources in predicting delphinid presence and illustrated how the integration of *in-situ* environmental data with acoustic data by WGs not only addresses the constraints of conventional surveys, but also enhances the identification of delphinid presence drivers and the identification of presence hot spots.

While autonomous platforms have emerged as a promising tool in marine science (*Klinck et al., 2012*; *Suberg et al., 2014*; *Klinck et al., 2016*; *Jones et al., 2019*; *Verfuss et al., 2019*; *Aniceto et al., 2020*; *Camus et al., 2021*), their application in cetacean monitoring has only begun to scratch the surface. Our study expands upon limited research that demonstrates the capability of these platforms to monitor odontocete species (*Bittencourt et al., 2017*; *Bittencourt et al., 2018*). Historically, odontocete populations have been monitored primarily through visual shipboard or aerial surveys, with acoustic monitoring largely restricted to stationary moored acoustic recording systems (*Wiggins & Hildebrand, 2007*; *Rafter et al., 2021*). Our study demonstrates the capacity of autonomous WGs to extend the reach of acoustic monitoring studies over greater distances, providing enhanced insights into the environmental variables that may predict odontocete presence. This, in turn, enhances our understanding of how these animals utilize shelf habitats in Florida waters.

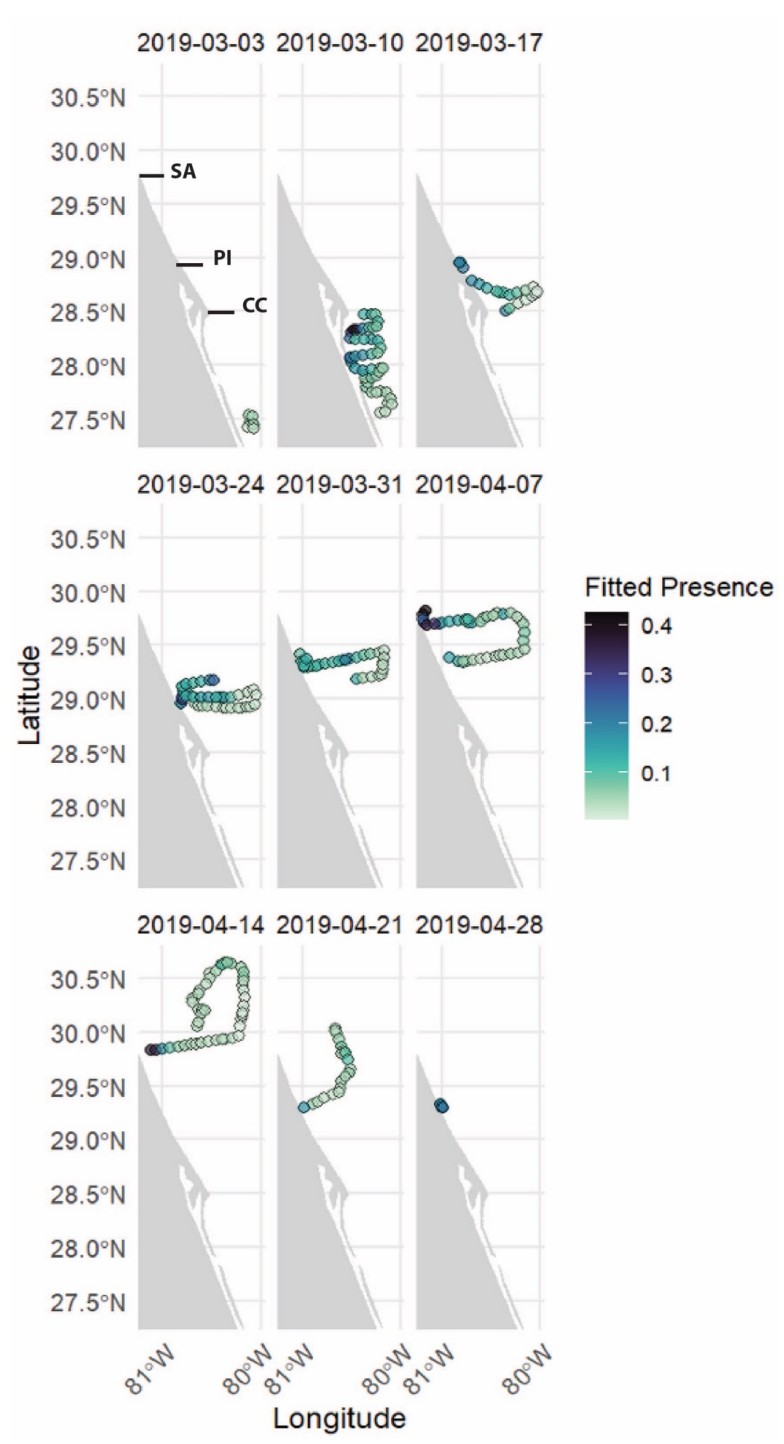

**Figure 4 Weekly maps of delphinid presence along the glider track as a function of date and location ID for the glider-only GAM (GOM).** Each image represents one week of the glider deployment, with the color scale indicating presence, with darker colors indicating higher detected presence and lighter colors indicating lower detected presence. SA, PI and CC indicate the location of St. Augustine, Ponce Inlet and Cape Canaveral, respectively.

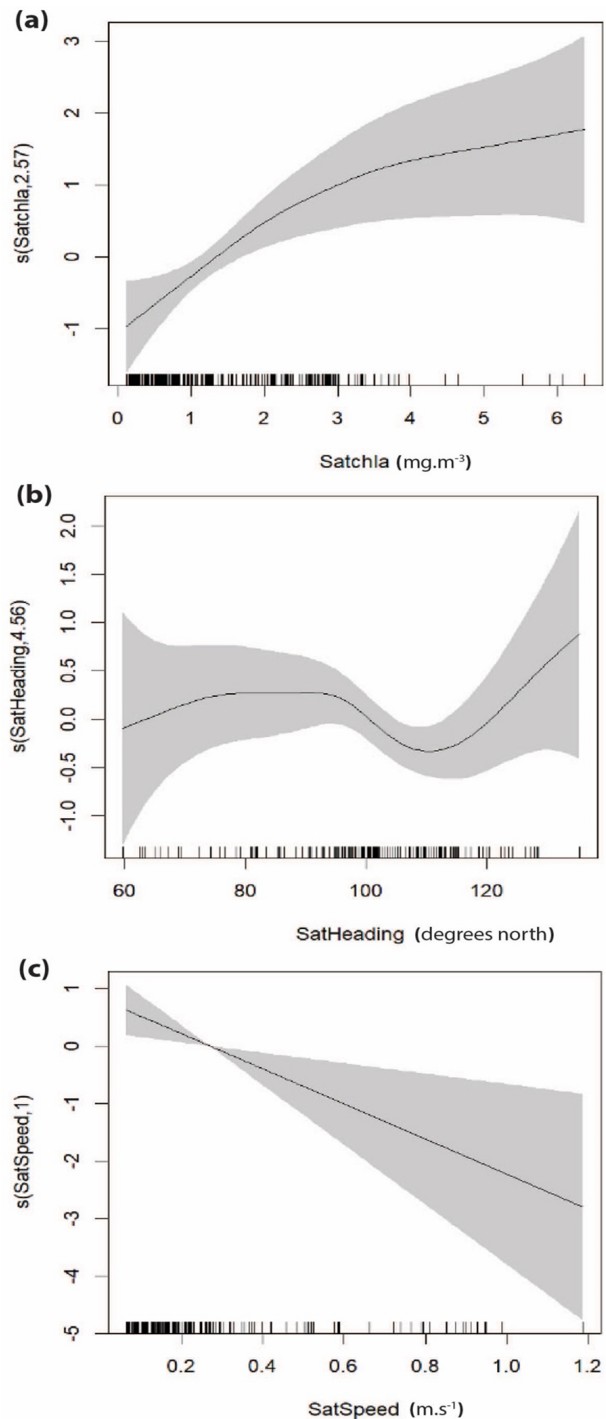

**Figure 5** **Smooth plots displaying the results of the top-performing satellite-only GAM (SOM) assessing the relationship between cetacean presence and environmental variables.** (A) Chlorophyll-a (mg m$^{-3}$); (B) current heading (degrees north); (C) current speed (m s$^{-1}$). 

**Figure 5 (…continued)**
Tick marks on the $x$-axis represent data points and gray shading represents the 95% confidence interval. The $y$-axis represents the relative effect of each variable on the presence of delphinids, with values above zero indicating increased presence and numbers below zero representing decreased presence. The number associated with each variable on the $y$-axis represents the effective degrees of freedom (EDF) for each smooth term, indicating the complexity of each variable's relationship with delphinid presence, 1 being the least complex.

Previous studies investigating environmental variables predicting cetacean presence have primarily relied on satellite-derived environmental data (*Roberts et al., 2016*; *Chavez-Rosales et al., 2019*; *Chavez-Rosales, Josephson & Garrison, 2022*). In this study, we performed a novel comparison between traditional cetacean monitoring methods, which utilize environmental data derived from satellites, and a newer approach employing autonomous WGs for *in situ* data collection. One of the primary findings of this study was the relevance of soundscape on the prediction of delphinid presence, with SPL included in the top three models of glider data. The addition of acoustic variables from the soundscape contributed to the superior performance of in situ-derived models in predicting delphinid presence compared to satellite-derived models. Although the difference in explained deviance between the GOM and SOM was not large, the ability to include aspects of the soundscape in analyses when using a glider is a clear advantage of this method. Without the addition of acoustic data, both GOM and SOM resulted in a similar explained deviance despite the difference in data sources.

The consistent inclusion of certain predictor variables across both *in situ* and satellite-derived models highlights their importance in shaping delphinid habitat preferences. Location and chlorophyll-a concentration emerged as important variables in all of the top performing models, indicating the importance of these variables as predictors of delphinid presence. The importance of location in all of the top performing models revealed several apparent hot spots for delphinid presence along the Florida Atlantic coast. The ability to identify these areas of interest, demonstrates another benefit of autonomous platforms. Furthermore, the inclusion of chlorophyll-a in all of the top models, aligns with previous research which has identified chlorophyll-a as an important predictor of cetacean presence for species such as minke whales (*Balaenoptera acutorostrata*), Risso's dolphin (*Grampus griseus*), white beaked dolphin (*Lagenorhynchus albirostris*), and several other dolphin species (*Breen, Brown & Rogan, 2016*; *Tardin et al., 2019*; *Sahri et al., 2021*). Increased odontocete presence with increased chlorophyll-a concentration may be due to an association with prey availability (*Lanz et al., 2009*; *Sachoemar, 2015*). Water temperature also emerged as an important predictor variable in nearly all models, with delphinid presence decreasing with higher temperatures, suggesting that the animals tended to avoid the Gulf Stream boundary. SPL was included in all three top GOMs, which highlights the influence that the soundscape has on these marine mammals, a factor not accounted for in traditional survey methods. Studies have demonstrated that anthropogenic noise exposure increases stress and elicits behavioral responses including diminished foraging in odontocete species such as bottlenose dolphins (*Tursiops truncatus*), orcas (*Orcinus orca*), harbour porpoises (*Phocoena phocoena*) and some arctic marine mammals (*Dyndo et al.,*

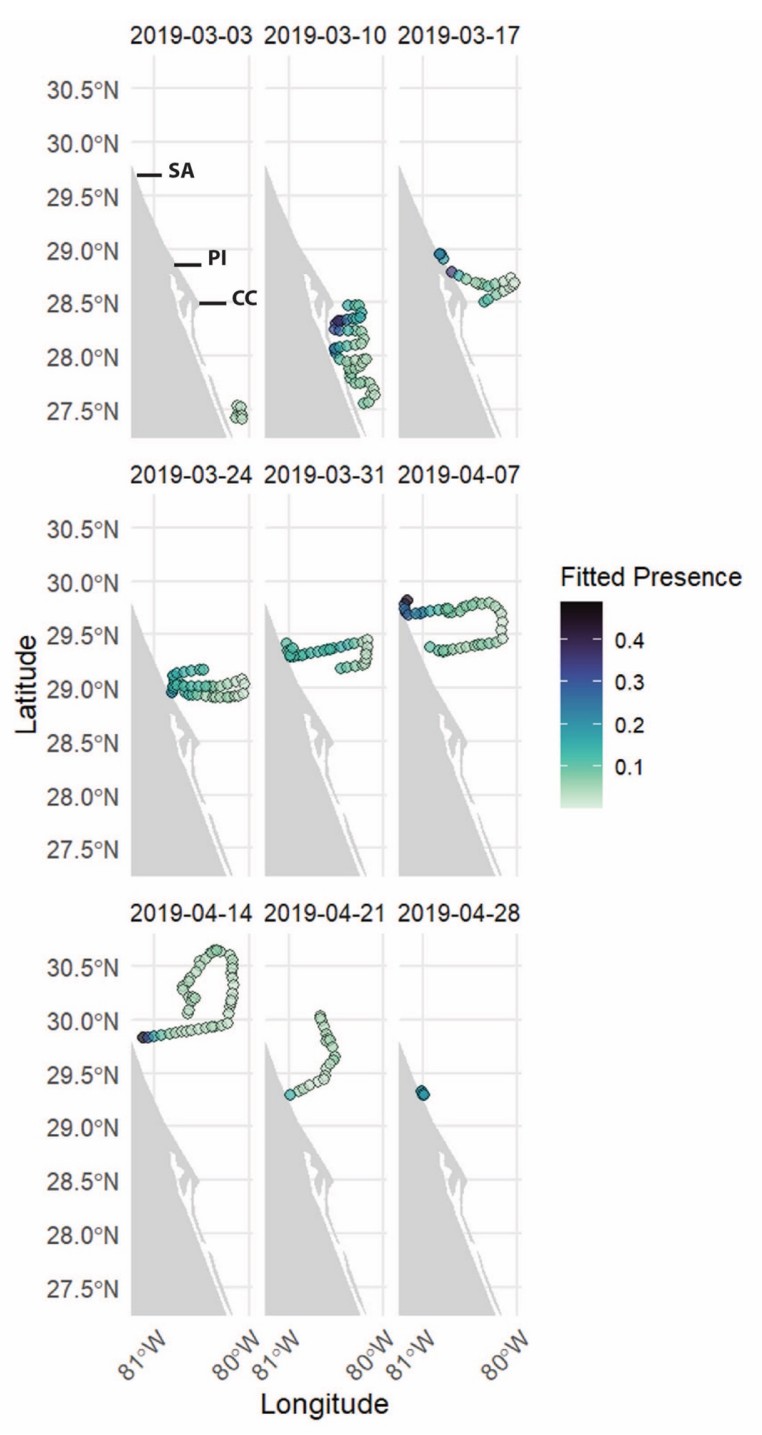

**Figure 6  Weekly maps of delphinid presence along the glider as a function of the date and location ID for the satellite-only GAM (SOM).** Each image represents one week of the glider deployment, with the color scale indicating presence, with darker colors indicating higher detected presence and lighter colors indicating lower detected presence.

*2015*; *Hauser & Stern, 2018*; *Yang et al., 2021*; *Martin et al., 2023*; *Tennessen et al., 2024*). Our findings suggest that delphinids avoid noisy environments and align with previous research emphasizing the role of environmental variables in shaping delphinid distribution patterns providing further support for their inclusion in habitat modeling efforts. It is worth noting, however, that the relationship between SPL and delphinid presence is complex. At mid-range SPLs and again at the highest SPL, there is a noticeable increase in delphinid presence (Fig. 3). This may indicate that delphinid species occur in and may even select somewhat noisy habitats. Such habitats, for example, may contain preferred prey such as near shore areas that typically have high SPL due to both natural (*e.g.*, wave action) and anthropogenic noise.

While the influence of chlorophyll-a and temperature on delphinid presence in our study aligned with previous research (*Breen, Brown & Rogan, 2016*; *Tardin et al., 2019*; *Sahri et al., 2021*), it was somewhat unexpected that depth was only included in one top model for both satellite and *in-situ* data. Depth has previously been shown to have a strong influence on several dolphin species, including bottlenose dolphins (*Tursiops truncatus*), primarily in neritic habitats, and striped dolphins (*Stenella coeruleoalba*), primarily in pelagic zones, across various regions such as the northeast Atlantic (*Breen, Brown & Rogan, 2016*) and the Gulf of Taranto in the Mediterranean (*Carlucci et al., 2016*). However, it is possible that this was due to limited variability of depths associated with the shelf environment and reduced sampling time in deeper offshore waters. Throughout the study area depth varied by 65 m. It is also possible that in the shelf environment other factors are more determinant for the presence of delphinids, such as food availability, which is usually found nearshore and in high chlorophyll-a concentration areas. *Scott et al. (2010)* analyzed the foraging habitats of seven species of marine apex predators observed simultaneously in a shallow sea, with continuous measurements taken of the detailed bio-physical water column characteristics to determine habitat preferences. All seven mobile top-predators preferentially foraged within habitats with small-scale (2 to 10 km) patches having (1) high concentrations of chlorophyll in the sub-surface chlorophyll maximum and (2) high variance in bottom topography, with different species preferring to forage in different locations within these habitats. *Scott et al. (2010)* showed that chlorophyll hotspots were associated with areas of locally increased vertical mixing. On the east Florida shelf such areas are located near inlets such as St. Augustine, Ponce Inlet, and also around Cape Canaveral where tidal mixing is significant, the three locations with the highest delphinid detection densities recorded in this study. *Blauw et al. (2012)* revealed how tides drive the variability of phytoplankton blooms. In the case of Cape Canaveral, export from inlets and rivers north of the Cape contribute to the significant amount of nutrient input found in this region where tidal flows are stronger than elsewhere on the East Florida Shelf, not including the inlets due to the bathymetric features and the shape of the cape (*Stelling et al., 2023*).

Our study continues to underscore the capabilities of autonomous gliders as a tool for cetacean monitoring and builds upon previous works in this field (*Klinck et al., 2012*; *Suberg et al., 2014*; *Bittencourt et al., 2018*; *Aniceto et al., 2020*; *Kowarski et al., 2020*; *Camus et al., 2021*). However, it is important to acknowledge some of the limitations associated with these methods. Primarily, acoustic monitoring is limited to when cetaceans are vocalizing.

Given that vocalizations are not continuous, and background noise can interfere, missed detections are possible. In this study recordings were made every 5 min, which increases the potential for missed detections. Furthermore, the difficulty in identifying species (particularly odontocetes) and inability to accurately estimate abundance are hurdles which acoustic monitoring has yet to overcome (*Mellinger et al., 2007*). Additional studies will be necessary to validate and expand upon our findings.

In conclusion, this study indicates that multiple factors influence cetacean distribution and abundance. Furthermore, it demonstrates the capability of autonomous gliders to simultaneously monitor delphinid species and their environments. We have shown that autonomous gliders can enrich our understanding of cetacean habitats by integrating environmental and PAM sensors that provide *in-situ* environmental and acoustic measurements simultaneously. Furthermore, we have demonstrated how gliders can offer deeper insights into delphinid presence compared to traditional monitoring methods, particularly with the addition of measurements within the soundscape, which are not captured in visual surveys and are limited in range with stationary acoustic monitoring. This study can also guide future research directions. For example, future studies should explore integrating glider data with visual surveys for more accurate species identification and investigate the interplay between cetaceans, their natural environment, and human influence.

## ACKNOWLEDGEMENTS

We would like to acknowledge Gabriel Alsenas and Erick Busold who handled the glider operation and Ali Altaher for helping with the acoustic data analysis. We would also like to acknowledge Dr. Steven Lombardo who helped refine and troubleshoot parameters for generalized additive models. We would also like to acknowledge the use of ChatGPT from OpenAI to troubleshoot coding scripts and for proofreading of small portions of the manuscript.

### Funding
This work was supported by the Harbor Branch Oceanographic Institute Foundation Specialty License Plates "Protect Wild Dolphins" and "Protect Florida Whales". The funders had no role in study design, data collection and analysis, decision to publish, or preparation of the manuscript.

### Grant Disclosures
The following grant information was disclosed by the authors:
The Harbor Branch Oceanographic Institute Foundation Specialty License Plates "Protect Wild Dolphins" and "Protect Florida Whales".

### Competing Interests
The authors declare there are no competing interests.

## Author Contributions

- Jessica Carvalho conceived and designed the experiments, analyzed the data, prepared figures and/or tables, authored or reviewed drafts of the article, and approved the final draft.
- Laurent M. Chérubin conceived and designed the experiments, performed the experiments, authored or reviewed drafts of the article, and approved the final draft.
- Greg O'Corry-Crowe conceived and designed the experiments, authored or reviewed drafts of the article, and approved the final draft.

## Data Availability

The raw data is available in the Supplementary Files.

## Supplemental Information

Supplemental information for this article can be found online at http://dx.doi.org/10.7717/peerj.19204#supplemental-information.

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

# PeerJ

**De Jong K, Forland TN, Amorim MCP, Rieucau G, Slabbekoorn H, Sivle LD. 2020.** Predicting the effects of anthropogenic noise on fish reproduction. *Reviews in Fish Biology and Fisheries* **30**:245–268 DOI 10.1007/s11160-020-09598-9.

**Di Tullio JC, Gandra TBR, Zerbini AN, Secchi ER. 2016.** Diversity and distribution patterns of cetaceans in the subtropical southwestern Atlantic outer continental shelf and slope. *PLOS ONE* **11**:e0155841 DOI 10.1371/journal.pone.0155841.

**Dyndo M, Wisniewska L, Rojano-Doñate P., Ryan D, Madsen T. 2015.** Harbour porpoises react to low levels of high frequency vessel noise. *Scientific Reports* **5**:11083 DOI 10.1038/srep11083.

**Elliser CR, Herzing DL. 2012.** Community structure and cluster definition of Atlantic spotted dolphins, *Stenella frontalis*, in the Bahamas. *Marine Mammal Science* **28**:E486–E502 DOI 10.1111/j.1748-7692.2012.00576.x.

**Erbe C, Marley SA, Schoeman RP, Smith JN, Trigg LE, Embling CB. 2019.** The Effects of Ship Noise on Marine Mammals–A Review. *Frontiers in Marine Science* **6**:606 DOI 10.3389/fmars.2019.00606.

**Forney KA. 2000.** Environmental models of cetacean abundance: reducing uncertainty in population trends. *Conservation Biology* **14**:1271–1286 DOI 10.1046/j.1523-1739.2000.99412.x.

**Gannon D, Waples D. 2004.** Diets of coastal bottlenose dolphins from the U.S. Mid-Atlantic coast differ by habitat. *Marine Mammal Science* **20**:527–545 DOI 10.1111/j.1748-7692.2004.tb01177.x.

**Garrison CAKL, Baumstark LI, Ward-Geiger R, Hines E. 2012.** Application of a habitat model to define calving habitat of the North Atlantic right whale in the southeastern United States. *Endangered Species Research* **18**:73–87 DOI 10.3354/esr00413.

**Garrison LP, Dias LA. 2023.** Abundance of Marine Mammals in Waters of the Southeastern U.S. Atlantic During Summer 2021. SEFSC MMTD Contribution: #MMTD-2023-01. Miami: Southeast Fisheries Science Center DOI 10.25923/ce0d-9e10.

**Garrison LP, Martinnez A, Maze-Foley K. 2010.** Habitat and abundance of cetaceans in Atlantic Ocean continental slope waters off the eastern USA. *Journal of Cetacean Research and Management* **11**:267–277 DOI 10.47536/jcrm.v11i3.606.

**GEBCO. 2023.** Gridded Bathymetry Data. International Hydrographic Organization (IHO) and the Intergovernmental Oceanographic Commission (IOC) of UNESCO. *Available at https://www.gebco.net/data_and_products/gridded_bathymetry_data/.*

**Gillespie D, Caillat M, Gordon J, White P. 2013.** Automatic detection and classification of odontocete whistles. *Journal of the Acoustical Society of America* **134(3)**:2427–2437 DOI 10.1121/1.4816555.

**Griffin R, Griffin NJ. 2004.** Temporal variation in Atlantic spotted dolphin (*Stenella frontalis*) and bottlenose dolphin (*Tursiops truncatus*) densities on the west Florida continental shelf. *Aquatic Mammals* **30**:380–390 DOI 10.1578/AM.30.3.2004.380.

**Häder D-P, Banaszak AT, Villafañe VE, Narvarte MA, González RA, Helbling EW. 2020.** Anthropogenic pollution of aquatic ecosystems: emerging problems with global implications. *Science of the Total Environment* **713**:136586 DOI 10.1016/j.scitotenv.2020.136586.

**Halpern BS, Selkoe KA, Micheli F, Kappel CV. 2007.** Evaluating and ranking the vulnerability of global marine ecosystems to anthropogenic threats. *Conservation Biology* **21**:1301–1315 DOI 10.1111/j.1523-1739.2007.00752.x.

**Halpern BS, Walbridge S, Selkoe KA, Kappel CV, Micheli F, D'Agrosa C, Bruno JF, Casey KS, Ebert C, Fox HE, Fujita R, Heinemann D, Lenihan HS, Madin EMP, Perry MT, Selig ER, Spalding M, Steneck R, Watson R. 2008.** A global map of human impact on marine ecosystems. *Science* **319**:948–952 DOI 10.1126/science.1149345.

**Haria SN, Hardy IC, Harzen S, Brunnick BJ. 2023.** Estimating population abundance of Atlantic bottlenose dolphins (*Tursiops truncatus*) in the coastal waters of Palm Beach County, southeastern Florida. *Aquatic Mammals* **49**:19–28 DOI 10.1578/AM.49.1.2023.19.

**Hauser DDW, Laidre KL, Stern HL. 2018.** Vulnerability of Arctic marine mammals to vessel traffic in the increasingly ice-free Northwest Passage and Northern Sea Route. *Proceedings of the National Academy of Sciences of the United States of America* **115**:7617–7622 DOI 10.1073/pnas.1803543115.

**Haver SM, Haxel J, Dziak RP, Roche L, Matsumoto H, Hvidsten C, Torres LG. 2023.** The variable influence of anthropogenic noise on summer season coastal underwater soundscapes near a port and marine reserve. *Marine Pollution Bulletin* **194(Pt A)**:115406 DOI 10.1016/j.marpolbul.2023.115406.

**Hawkins ER, Gartside DF. 2009.** Interactive behaviours of bottlenose dolphins (*Tursiops aduncus*) during encounters with vessels. *Aquatic Mammals* **35**:259–268 DOI 10.1578/AM.35.2.2009.259.

**Herzing DL, Elliser CR. 2016.** Opportunistic sightings of cetaceans in nearshore and offshore waters of southeast Florida. *Journal of Northwest Atlantic Fishery Science* **48**:21–31 DOI 10.2960/J.v48.m709.

**Hildebrand JA, Baumann-Pickering S, Frasier KE, Trickey JS, Merkens KP, Wiggins SM, McDonald MA, Garrison LP, Harris D, Marques TA, Thomas L. 2015.** Passive acoustic monitoring of beaked whale densities in the Gulf of Mexico. *Scientific Reports* **5**:16343 DOI 10.1038/srep16343.

**Jensen FH, Bejder L, Wahlberg M, Aguilar Soto N, Johnson M, Madsen PT. 2009.** Vessel noise effects on delphinid communication. *Marine Ecology Progress Series* **395**:161–175 DOI 10.3354/meps08204.

**Jones DOB, Gates AR, Huvenne VAI, Phillips AB, Bett BJ. 2019.** Autonomous marine environmental monitoring: application in decommissioned oil fields. *Science of the Total Environment* **668**:835–853 DOI 10.1016/j.scitotenv.2019.02.310.

**Keller CA, Ward-Geiger LI, Brooks WB, Slay CK, Taylor CR, Zoodsma BJ. 2006.** North Atlantic right whale distribution in relation to sea-surface temperature in the southeastern United States calving grounds. *Marine Mammal Science* **22**:426–445 DOI 10.1111/j.1748-7692.2006.00033.x.

**Kiszka JJ, Woodstock MS, Heithaus MR. 2022.** Functional roles and ecological importance of small cetaceans in aquatic ecosystems. *Frontiers in Marine Science* **9**:803173 DOI 10.3389/fmars.2022.803173.

**Klinck H, Fregosi S, Matsumoto H, Turpin A, Mellinger DK, Erofeev A, Barth JA, Shearman RK, Jafarmadar K, Stelzer R. 2016.** Mobile autonomous platforms for passive-acoustic monitoring of high-frequency cetaceans. In: *Robotic Sailing 2015.* Cham: Springer International Publishing, 39–37.

**Klinck H, Mellinger DK, Klinck K, Bogue NM, Luby JC, Jump WA, Shilling GB, Litchendorf T, Wood AS, Schorr GS, Baird RW. 2012.** Near-real-time acoustic monitoring of beaked whales and other cetaceans using a Seaglider™. *PLOS ONE* **7**:e36128 DOI 10.1371/journal.pone.0036128.

**Kowarski KA, Gaudet BJ, Cole AJ, Maxner EE, Turner SP, Martin SB, Johnson HD, Moloney JE. 2020.** Near real-time marine mammal monitoring from gliders: practical challenges, system development, and management implications. *The Journal of the Acoustical Society of America* **148**:1215–1230 DOI 10.1121/10.0001811.

**Lanz E, Martinez J, Nevarez Martinez M, Dworak JA. 2009.** Small pelagic fish catches in the Gulf of California associated with sea surface temperature and chlorophyll. *California Cooperative Oceanic Fisheries Investigations Reports* **50**:134–146.

**Malinka CE, Gillespie DM, Macaulay JD, Joy R, Sparling CE. 2018.** First in situ passive acoustic monitoring for marine mammals during operation of a tidal turbine in Ramsey Sound, Wales. *Marine Ecology Progress Series* **590**:247–266 DOI 10.3354/meps12467.

**Marley SA, Salgado Kent CP, Erbe C, Parnum IM. 2017.** Effects of vessel traffic and underwater noise on the movement, behaviour and vocalisations of bottlenose dolphins in an urbanised estuary. *Scientific Reports* **7**:13437 DOI 10.1038/s41598-017-13252-z.

**Martin MJ, Halliday WD, Storrie L, Citta JJ, Dawson J, Hussey NE, Juanes F, Loseto LL, MacPhee SA, Moore L, Nicoll A, O'Corry-Crowe G, Insley SJ. 2023.** Exposure and behavioral responses of tagged beluga whales (*Delphinapterus leucas*) to ships in the Pacific Arctic. *Marine Mammal Science* **39**:387–421 DOI 10.1111/mms.12978.

**Mathworks I. 2022.** MATLAB Version: 9.13.0 (R2022b). Natick: MathsWorks, Inc.

**Mazzoil M, Gibson Q, Durden WN, Borkowski R, Biedenbach G, McKenna Z, Gordon N, Brightwell K, Denny M, Howells E. 2020.** Spatiotemporal movements of common bottlenose dolphins (*Tursiops truncatus truncatus*) in Northeast Florida, USA. *Aquatic Mammals* **46(3)**:285–300 DOI 10.1578/AM.46.3.2020.285.

**Mazzoil M, Murdoch ME, Reif JS, Bechdel SE, Howells E, De Sieyes M, Lawrence C, Bossart GD, McCulloch SD. 2011.** Site fidelity and movement of bottlenose dolphins (*Tursiops truncatus*) on Florida's east coast: atlantic Ocean and Indian River Lagoon Estuary. *Florida Scientist* 25–37.

**Mellinger DK, Stafford KM, Moore SE, Dziak RP, Matsumoto H. 2007.** An overview of fixed passive acoustic observation methods for cetaceans. *Oceanography* **20**:36–45 DOI 10.5670/oceanog.2007.03.

**Merchant ND, Fristrup KM, Johnson MP, Tyack PL, Witt MJ, Blondel P, Parks SE. 2015.** Measuring acoustic habitats. *Methods in Ecology and Evolution* **6**:257–265 DOI 10.1111/2041-210X.12330.

**Miller DL, Burt ML, Rexstad EA, Thomas L, Gimenez O. 2013.** Spatial models for distance sampling data: recent developments and future directions. *Methods in Ecology and Evolution* **4**:1001–1010 DOI 10.1111/2041-210X.12105.

**NOAA CoastWatch. 1997.** NOAA CoastWatch Near Real-time Ocean Color Sea-Viewing Wide Field-of-View Sensor (SeaWiFS) Products. Sea surface temperature, multi-scale ultra-high resolution (MUR JPL), Daily 1 km East Coast EEZ, 2003-2021. In: T. Lab JP (ed). Washington, D.C: NOAA National Centers for Environmental Information. Dataset. *Available at https://www.ncei.noaa.gov/archive/accession/CoastWatch-OC-SeaWiFS*.

**Popper AN, Hawkins AD. 2019.** An overview of fish bioacoustics and the impacts of anthropogenic sounds on fishes. *Journal of Fish Biology* **94**:692–713 DOI 10.1111/jfb.13948.

**Quintana-Rizzo E, Mann DA, Wells RS. 2006.** Estimated communication range of social sounds used by bottlenose dolphins (*Tursiops truncatus*). *Journal of the Acoustical Society of America* **120**:1671–1683 DOI 10.1121/1.2226559.

**Rafter MA, Rice AC, Berga AS, Frasier KE, Thayre BJ, Majewski D, Wiggins SM, Baumann-Pickering S, Hildebrand JA. 2021.** Passive Acoustic Monitoring for Marine Mammals in the Jacksonville Range Complex June 2019–June 2020. Final Report. Marine Physical Laboratory Technical Memorandum 656. December 2021. Submitted to Naval Facilities Engineering Systems Command (NAVFAC) Atlantic, Norfolk, Virginia, under Contract No. N62470-15-D-8006 Subcontract #383-8476 (MSA2015-1176 Task Order 003) issued to HDR, Inc.

**Richardson WJ, Würsig B. 1997.** Influences of man-made noise and other human actions on cetacean behaviour. *Marine & Freshwater Behaviour & Phy* **29**:183–209 DOI 10.1080/10236249709379006.

**Rickard M. 2015.** A spatio-temporal gap analysis of cetacean survey effort in the U.S. Mid- and South Atlantic. Master's thesis, Duke University, Durham, NC, USA *Available at https://hdl.handle.net/10161/9679*.

**Roberts JJ, Best BD, Mannocci L, Fujioka E, Halpin PN, Palka DL, Garrison LP, Mullin KD, Cole TVN, Khan CB, McLellan WA, Pabst DA, Lockhart GG. 2016.** Habitat-based cetacean density models for the U.S. Atlantic and Gulf of Mexico. *Scientific Reports* **6**:22615 DOI 10.1038/srep22615.

**Roch MA, Klinck H, Baumann-Pickering S, Mellinger DK, Qui S, Soldevilla MS, Hildebrand JA. 2011.** Classification of echolocation clicks from odontocetes in the Southern California Bight. *Journal of the Acoustical Society of America* **129**(1):467–475 DOI 10.1121/1.3514383.

**Sachoemar S. 2015.** Variability of sea surface chlorophyll-a, temperature and fish catch within Indonesian region revealed by satellite data. *Marine Research in Indonesia* **37**:75–87 DOI 10.14203/mri.v37i2.25.

**Sahri A, Herwata Putra MI, Kusuma Mustika PL, Kreb D, Murk AJ. 2021.** Cetacean habitat modelling to inform conservation management, marine spatial planning, and as a basis for anthropogenic threat mitigation in Indonesia. *Ocean & Coastal Management* **205**:105555 DOI 10.1016/j.ocecoaman.2021.105555.

**Scott BE, Sharples J, Ross ON, Wang J, Pierce GJ, Camphuysen CJ. 2010.** Sub-surface hotspots in shallow seas: fine-scale limited locations of top predator foraging habitat indicated by tidal mixing and sub-surface chlorophyll. *Marine Ecology Progress Series* **408**:207–226 DOI 10.3354/meps08552.

**Simpson SD, Radford AN, Nedelec SL, Ferrari MCO, Chivers DP, McCormick MI, Meekan MG. 2016.** Anthropogenic noise increases fish mortality by predation. *Nature Communications* **7**:10544 DOI 10.1038/ncomms10544.

**Soldevilla MS, Elizabeth Henderson E, Campbell GS, Wiggins SM, Hildebrand JA, Roch JA. 2008.** Classification of Risso's and Pacific white-sided dolphins using spectral properties of echolocation clicks. *Journal of the Acoustical Society of America* **124(1)**:609–624 DOI 10.1121/1.2932059.

**Soldevilla MS, Wiggins SM, Hildebrand JA. 2010.** Spatial and temporal patterns of Risso's dolphin echolocation in the Southern California Bight. *The Journal of the Acoustical Society of America* **127**:124–132 DOI 10.1121/1.3257586.

**Sousa-Lima RS, Fernandes DP, Norris TF, Oswald JN. 2013.** A review and inventory of fixed autonomous recorders for passive acoustic monitoring of marine mammals: 2013 state-of-the-industry. In: *2013 IEEE/OES acoustics in underwater geosciences symposium*. Piscataway: IEEE, 1–9.

**Stelling B, Phlips S, Badylak L, Landauer M, Tate E, West-Valle A. 2023.** Seasonality of phytoplankton biomass and composition on the Cape Canaveral shelf of Florida: role of shifts in climate and coastal watershed influences. *Frontiers in Ecology and Evolution* **11**:1134069 DOI 10.3389/fevo.2023.1134069 .

**Suberg L, Wynn RB, Kooij Jvd, Fernand L, Fielding S, Guihen D, Gillespie D, Johnson M, Gkikopoulou KC, Allan IJ, Vrana B, Miller PI, Smeed D, Jones AR. 2014.** Assessing the potential of autonomous submarine gliders for ecosystem monitoring across multiple trophic levels (plankton to cetaceans) and pollutants in shallow shelf seas. *Methods in Oceanography* **10**:70–89 DOI 10.1016/j.mio.2014.06.002.

**Tardin RH, Chun Y, Jenkins CN, Maciel IS, Simão SM, Alves MAS. 2019.** Environment and anthropogenic activities influence cetacean habitat use in southeastern Brazil. *Marine Ecology Progress Series* **616**:197–210 DOI 10.3354/meps12937.

**Tennessen JB, Holt MM, Wright BM, Hanson MB, Emmons CK, Giles DA, Hogan JT, Thornton SJ, Deecke VB. 2024.** Males miss and females forgo: auditory masking from vessel noise impairs foraging efficiency and success in killer whales. *Global Change Biology* **30**:e17490 DOI 10.1111/gcb.17490.

**Thomisch K, Boebel O, Clark CW, Hagen W, Spiesecke S, Zitterbart DP, Van Opzeeland I. 2016.** Spatio-temporal patterns in acoustic presence and distribution of Antarctic blue whales Balaenoptera musculus intermedia in the Weddell Sea. *Endangered Species Research* **30**:239–253 DOI 10.3354/esr00739.

**Verfuss UK, Aniceto AS, Harris DV, Gillespie D, Fielding S, Jiménez G, Johnston P, Sinclair RR, Sivertsen A, Solbø SA, Storvold R, Biuw M, Wyatt R. 2019.** A review of unmanned vehicles for the detection and monitoring of marine fauna. *Marine Pollution Bulletin* **140**:17–29 DOI 10.1016/j.marpolbul.2019.01.009.

**Weilgart LS. 2007.** The impacts of anthropogenic ocean noise on cetaceans and implications for management. *Canadian Journal of Zoology* **85**:1091–1116 DOI 10.1139/Z07-101.

**Wiggins SM, Hildebrand JA. 2007.** High-frequency Acoustic Recording Package (HARP) for broad-band, long-term marine mammal monitoring. In: *2007 symposium on underwater technology and workshop on scientific use of submarine cables and related technologies.* Piscataway: IEEE, 551–557.

**Yang W-C, Chen C-F, Chuah Y-C, Zhuang C-R, Chen I-H, Mooney TA, Stott J, Blanchard M, Jen I-F, Chou L-S. 2021.** Anthropogenic sound exposure-induced stress in captive dolphins and implications for cetacean health. *Frontiers in Marine Science* **8**:606736 DOI 10.3389/fmars.2021.606736.