# Peer review of "Autonomous wave gliders as a tool to characterize delphinid habitats along the Florida Atlantic coast"

_PeerJ, doi:10.7717/peerj.19204_

## Round 0.1 · original submission · Minor Revisions

I believe this is a valuable manuscript, which is reflected in the mostly positive comments from two reviewers. Please address their comments, with an emphasis on the methods/results and considering which species to include. There are also a few items with figures to address. In addition to reviewer suggestions, I would also add subpanel labels to make the captions more clear.

Reviewer 1 ·

Basic reporting

The study is very interesting, in particulat the use of innovative methodology like the ocean gliders. The paper is very well written and very clear.

Experimental design

I have some doubts of the data collection. In my opinion it would be better consider only delphinids. In the abstract you mentioned you detect mostly them, I would remove from the dataset deep diver such as sperm whales and beaked whales (if you detect any). Try to run the models again and maybe you can have more robust results.

Validity of the findings

Tha analyses are well described. I would just try to keep only the delphinids in the dataset and see how the results change.

Additional comments

Introduction
Lines 22-35 and lines 45-53: I would merge these 2 sections with a section speaking what odontocetes species are common in the area with a short description on their ecology. Are there sperm whales or beaked whales in the area? Maybe you can add a table with the odontocetes you could find in the area and their habitat so in your results and discussion you have something to compare your findings. If you are considering bottlenose dolphins (inshore) and sperm whale (offshore) in the same dataset it is very possible that the distance from shore is not a relevant component.
Lines 82-84: I would add a reference here. I would also expand this part. In your introduction you don’t speak much about the anthropogenic noise or the sound pressure level while in your discussion you are using it as one of the main topics.
Line 92: I would use the term “vehicle” instead of “platform”.

Methods
Line 127: I would change a bit Figure 2. Maybe you can create a more squared image than this one.
Line 142: I would add the specifics of the hydrophone (sensitivity etc.)
Lines 157-158: Here I would specify a bit more what parameters you choose. Echolocation clicks from beaked whales and sperm whales are different than the ones from small delphinids. Moreover, with your sampling frequency I am not sure you could get the beaked whale ones. Did you find also sperm whale clicks during your study? Otherwise, you can just say you were checking for delphinids vocalizations.
Lines 183-184: How did you identify visually the acoustic presence of fish? Did you check for some particular vocalizations? At what frequencies ranges? Same for the anthropogenic noise. Did you use the same spectrogram settings to detect anthropogenic noise, fish, odontocetes? Is there a possibility to not distinguish (at low frequencies especially) between the anthropogenic noise and the noise from the glider?
Line 184: I am not sure about the fish presence. What species in particular you were looking for? Are preys for what species of dolphin? I would specify here a bit more with some references to justify this data collected or I would just remove from the research as it is not part of your model.
Line 194: I would add the version of QGIS and R Studio.
Line 195: I would add the software and the version used for the statistical analyses.

Results
Lines 215-217: Did you detect sperm whales? Or only delphinids? I would specify it here.
Line 227: In Figure 3 the unity of measure in x-axis is missing.
Lines 235-240: I would put this figure bigger, adding in the figure the location of the areas you are mentioning in the results (Cape Canaveral and St. Augustin inlet).
Lines 249-250: Here you are mentioning the distance from the shore but in your model the distant from the shore seems to be not relevant. Why do you think the location (and not distance from the shore) is important in your results?

Discussion
I would discuss more about the possible species you are detecting. What are the odontocetes species more common in the area? Do you think you are getting some particular results because of the mix of different species? I would add more about the ecology of the odontocetes you are considering in the study.
Lines 314-315: There are many studies about this topic. I would cite here something more generic and not specific only for bottlenose dolphins.
Line 327: In my opinion considering all the odontocetes you are taking into account many species that live in very different habitat. Sperm whales prefer deep waters and slope, offshore dolphin species prefer deep waters, bottlenose dolphins stay close to the shore.

Reviewer 2 ·

Basic reporting

The work "Autonomous wave gliders as a tool to characterize odontocete habitats along the Florida Atlantic coast" is really very interesting and deals with a topic that will see, in the near future, a significant development.
The use of drones, autonomous vehicles and in general of advanced technological tools to support environmental monitoring of habitats and environments is the future.
the study area is also one of the historically monitored areas and often an example for works all over the world.
The work is quite fluent, perhaps a little long the introduction that sees 5 pages before reaching the purpose of the work itself. Certainly the authors will be better able to dry the concepts and studies previously carried out to indicate the focus of the work.

Experimental design

The methods should be better detailed for the selection procedure of whistles and clicks and the exclusion of other sounds such as buzzes, burst-pulses should be specified. Furthermore, it is not clear whether the recorded species are recognized or not and this makes the difference in the discussion of some results.

Validity of the findings

In fact, sometimes in the discussions of the results obtained and the relationships between cetacean presence/absence and parameters it is clear that having used all the vocalizations (without distinction of species) leads to a presence/absence-variable relationship that seems contrasting.
This behavior could be due to the assembly of data relating to different species that have different acoustic and distribution behaviors.
Overall, however, the work deserves publication, after revision.

Annotated reviews are not available for download in order to protect the identity of reviewers who chose to remain anonymous.

---

## Round 0.2 · accepted · Accept

Thank you for your revised manuscript. I appreciate you fully addressing reviewer comments. I am happy with this current version after reviewing it myself. I believe it is now ready for publication.